# Microbial Transformation of Licochalcones

**DOI:** 10.3390/molecules25010060

**Published:** 2019-12-23

**Authors:** Yina Xiao, Fubo Han, Ik-Soo Lee

**Affiliations:** College of Pharmacy, Chonnam National University, Gwangju 61186, Korea

**Keywords:** licorice, licochalcone, retrochalcone, fungal transformation.

## Abstract

Microbial transformation of licochalcones B (**1**), C (**2**), D (**3**), and H (**4**) using the filamentous fungi *Aspergillus niger* and *Mucor hiemalis* was investigated. Fungal transformation of the licochalcones followed by chromatographic separations led to the isolation of ten new compounds **5**–**14**, including one hydrogenated, three dihydroxylated, three expoxidized, and three glucosylated metabolites. Their structures were elucidated by combined analyses of UV, IR, MS, NMR, and CD spectroscopic data. Absolute configurations of the 2″,3″-diols in the three dihydroxylated metabolites were determined by ECD experiments according to the Snatzke’s method. The *trans*-*cis* isomerization was observed for the metabolites **7**, **11**, **13**, and **14** as evidenced by the analysis of their ^1^H-NMR spectra and HPLC chromatograms. This could be useful in better understanding of the *trans*-*cis* isomerization mechanism of retrochalcones. The fungal transformation described herein also provides an effective method to expand the structural diversity of retrochalcones for further biological studies.

## 1. Introduction

Licorice is one of the oldest and most commonly used herbal medicines in the world [1]. The name ‘licorice’ is derived from the dried roots of *Glycyrrhiza* species (Leguminosae family) and is the common name for these plants [2]. A variety of pharmacological activities have been described for licorice, and these bioactivities are attributed to the chemical constituents of licorice [3]. To date, more than 600 compounds have been isolated from licorice, and the major active constituents are saponins and flavonoids [4]. Among them, a series of retrochalcones, licochalcones A, B, C, D, E, G and the closely related compound echinatin, were identified from the roots of licorice [5]. Structurally, these retrochalcones belong to an unusual phenolic compound family and are distinguished from ordinary chalcones by the lack of a hydroxyl group at the C-2′ and C-6′ positions [6]. Several biological mechanism studies have proved that these compounds exhibited diverse biological effects such as anti-oxidant [7], antibacterial [8], anti-inflammatory [9,10], anticancer [11,12,13,14], osteogenesis [15], anti-hepatotoxic, antidiabetic and anti-allergic effects [16]. Moreover, the novel synthetic licochalcone F, a regioisomer of licochalcone E, showed in vivo anti-inflammatory and glucose intolerance effects without side effects [17]. Another synthetic licochalcone H, a regioisomer of licochalcone C, was found to exhibit anti-cancer activity by promoting apoptosis in human esophageal squamous cell carcinoma cells [18]. Even though promising biological studies of these small molecules were investigated, it is still essential to explore and generate more novel candidates for possible therapeutic uses [19].

Microbial transformation by filamentous fungi can be used as a feasible alternative to the conventional chemical methods when searching for new derivatives with potential biological properties [20]. These fungi could improve the regio- and stereo-selectivity of some chemical reactions [21]. The application of microbes for biotransformation of chalcones led to the formation of many novel derivatives by means of cyclization, hydroxylation, reduction, and dehydrogenation reactions [22,23,24]. However, to date few microbial transformation studies have been investigated directly on the series of retrochalcones isolated from licorice. In the present study, we report the microbial transformation of licochalcones B (LB, **1**), C (LC, **2**), D (LD, **3**) and H (LH, **4**) by the selected fungi *Aspergillus niger* and *Mucor hiemalis*.

## 2. Results and Discussion

### 2.1. Microbial Transformation of Licochalcones B *(**1**)*, C *(**2**)*, D *(**3**)* and H *(**4**)* by A. niger

Microbial transformation of LB (**1**) by A. niger produced the hydrogenated metabolite **5**; microbial transformation of LC (**2**), LD (**3**) and LH (**4**) by A. niger furnished the corresponding dihydroxylated metabolites **6**, **8**, **10** and expoxidized metabolites **7**, **9**, **11**, respectively (Scheme 1).

Metabolite **5** was obtained as a yellow amorphous powder. Its molecular formula was determined as C_16_H_16_O_5_ by the positive [M + Na]^+^ peak at *m/z* 311.0896 (calcd for C_16_O_16_O_5_Na, 311.0895) on the basis of its HRESIMS spectrum, which indicated that it was a dihydrogenerated derivative of **1**. In the ^1^H-NMR spectrum of **5**, the trans-olefinic proton signals of H-α and H-β were replaced by two new methylene proton signals at δ_H_ 3.15 (2H, t, *J* = 7.4 Hz) and 2.87 (2H, t, *J* = 7.4 Hz). Consistently, ^13^C-NMR spectrum of **5** showed the absence of low field olefinic carbon signals and the presence of two new high field carbon signals at δ_C_ 39.2 and 25.1. Location of the hydrogenation was further determined to be at C-α and C-β based on the correlations of H-α (δ_H_ 3.15) and H-β (δ_H_ 2.87) with C=O (δ_C_ 200.0) in the HMBC spectrum of **5**. Thus, coupound **5** was established to be α,β-dihydrolicochalcone B.

Microbial transformation of licochalcone C (**2**) by *A. niger* for three days produced metabolites **6** and **7**. They were obtained as yellow amorphous powders. Metabolite **6** gave a molecular formula of C_21_H_24_O_6_ from its HRESIMS, indicating that two oxygen atoms and two hydrogen atoms were inserted into the substrate **2**. UV spectrum of **6** showed characteristic absorption maxima at 207 and 352 nm, indicating that the chalcone skeleton remained intact. The ^1^H-NMR spectrum of **6** showed two doublet proton signals at δ_H_ 8.00 and 7.63 (each d, *J* = 15.6 Hz), which are characteristic of the *E*-configuration of the α,β-double bond in the chalcone derivatives. In comparison with the ^1^H-NMR spectrum of **2**, two proton signals H-1″ (δ_H_ 3.45, 2H, d, *J* = 6.9 Hz) and H-2” (δ_H_ 5.23, m, 1H) on the prenyl moiety disappeared, whereas three new proton signals δ_H_ 2.69 (1H, dd, *J* = 14.2, 10.3 Hz), 3.12 (1H, dd, *J* = 14.2, 2.0 Hz) and 3.65 (1H, dd, *J* = 10.3, 2.0 Hz) were observed in **6**. It was suggested that two hydroxyl groups were newly introduced at C-2″ and C-3″. Dihydroxylation was also supported by ^13^C-NMR based on the presence of two new oxygen-bearing ^13^C signals at δ_C_ 72.6 (C-3″) and 78.9 (C-2″) instead of two olefinic carbon signals [24]. In HMBC spectrum, the proton signals at δ_H_ 2.69 (H-1″b) and 3.12 (H-1″a) were correlated with the carbon signal at δ_C_ 120.9 (C-3), and the proton signal at δ_H_ 3.65 (H-2″) was correlated with the carbon signal at δ_C_ 72.6 (C-3″). Absolute configuration of the 2″,3″-diols in **6** was determined using the ECD spectrum obtained by the Snatzke’s method [25,26,27]. Based on the empirical rule, induced CD (ICD) curve of the Mo-complex of **6** showed the negative Cotton effect at around 310 nm, indicating that the O-C-C-O dihedral angle in the favored conformation was negative for **6** (Figure 1). Therefore, the absolute configuration at C-2″ of **6** was assigned *R*. Based on these data, the structure of **6** was elucidated as (2″*R*)-2″,3″-dihydroxylicochalcone C.

The molecular formula of the metabolite **7** was revealed as C_21_H_22_O_5_ based on the HRESIMS ion peak at 377.1376 (calcd for C_21_O_22_O_5_Na, 377.1365), indicating the addition of one oxygen atom compared to the substrate **2**. Interestingly, the HPLC chromatogram of the purified **7** showed two peaks at *t*_R_ of 22.66 and 26.06 min. The corresponding UV λ_max_ of the two peaks were 240, 357 nm and 293 nm, respectively. Moreover, the ^1^H-NMR peak pattern of **7** showed two sets of resonance signals in a ratio of approximately 1:0.3. They were clearly assignable to the suggested major (*E*)- and minor (*Z*)-isomers by observing the two sets of H-α and H-β proton signals including those at δ_H_ 7.64 (1H, d, *J* = 15.6 Hz) and 8.01 (1H, d, *J* = 15.6 Hz) in one set, and those at δ_H_ 6.55 (1H, d, *J* = 12.8 Hz) and 7.11 (1H, d, *J* = 12.8 Hz) in another. Assignment of the major *trans* isomer was confirmed by the presence of three new proton signals at δ_H_ 3.80 (1H, dd, *J* = 7.0, 5.0 Hz, H-2″), 3.04 (1H, dd, *J* = 17.1, 5.0 Hz, H-1″a) and 2.73 (1H, dd, *J* = 17.1, 7.0 Hz, H-1″b) coupled with the ^13^C-NMR signals at δ_c_ 68.3 (C-2″) and 25.8 (C-1″) based on the NMR analyses of **7**. Location of the epoxidation of **7** was confirmed to be at C-2″ and C-3″ according to the HMBC correlation from H-1″a and H-1″b to the additional quaternary carbon signal at δ_c_ 77.3 (C-3″). Minor *cis*-form showed the same correlation patterns comparing with those of *trans* set signals in the 1D and 2D NMR spectra. Moreover, the assignments of the proton signals revealed general up-field shifts in the *cis* isomer as compared to those of the *trans* isomer in the ^1^H-NMR spectrum. Therefore, the structure of **7** was confirmed as 2″, 3″-expoxylicochalcone C.

Microbial transformation of licochalcone D (**3**) by *A. niger* for three days produced the metabolites **8** and **9**. Compounds **8** and **9** were obtained as yellow amorphous powders. The molecular formula of **8** (C_21_H_24_O_7_) was compatible with the insertion of two oxygen and two hydrogen atoms to the substrate **3**, which was supported by the analysis of its HRESIMS data (*m/z* 389.1603 [M + H]^+^, calcd for C_21_H_25_O_7_, 389.1600). The ^1^H-NMR and ^13^C-NMR spectroscopic data of **8** were almost identical with those of **3** except for resonances of the prenyl moiety. Presence of the prenyl moiety gave rise to the proton signals at δ_H_ 2.60 (1H, dd, *J* = 13.8, 10.4 Hz), 3.13 (1H, dd, *J* = 13.8, 1.3 Hz) and 3.66 (1H, dd, *J* = 10.4, 1.3 Hz), and two oxygen-bearing carbon signals at δ_C_ 78.1 and 72.5 instead of original olefinic signals. The locations of dihydroxylation were confirmed to be at C-2″ and C-3″ by the HMBC correlations from H-4″ and H-5″ to C-2″ and C-3″, and from H-1″ to C-2″ and C-3″. The absolute configuration at C-2″ was determined by comparison of their ICD spectra using the Snatzke’s method. The Mo-complex of compound **8** presented a negative Cotton effect at around 310 nm (Appendix A), indicating the *R* configuration at C-2″. Therefore, metabolite **8** was identified as (2″*R*)-2″,3″-dihydroxylicochalcone D.

Compound **9** had a molecular formula of C_21_H_22_O_6_ on the basis of its HRESIMS data (*m/z* 371.1497 [M + H]^+^, calcd for C_21_H_23_O_6_, 371.1495) which lacks H_2_O moiety compared with that of **8**. The NMR data of **9** were similar to those of **8**, implying **9** possessed the same skeleton as **8**. However, major differences in their ^1^H-NMR spectra were observed in the resonances of δ_H_ 3.82 (1H, dd, *J* = 7.0, 5.0, H-2″), 3.11 (1H, dd, *J* = 16.8, 5.0, H-1″a), 2.82 (1H, dd, *J* = 16.8, 7.0, H-1″b), 1.36 (3H, s, H-4″), 1.31 (3H, s, H-5″), suggesting that the differences between **8** and **9** were at C-2″ and C-3″. In the HMBC spectrum, the H-1″a and H-1″b were coupled with the oxygen-bearing carbon signals at δ_C_ 68.6 (C-2″) indicating the epoxidation of the double bond in the prneyl group. Moreover, the proton signals H-4″ (δ_H_ 1.36) and H-5″ (δ_H_ 1.31) showed correlations with C-2″ and C-3″. Thus, metabolite **9** was identified as 2″,3″-expoxylicochalcone D.

Microbial transformation of licochalcone H (**4**) by *A. niger* for three days produced metabolites **10** and **11**. The molecular formula of metabolite **10**, C_21_H_24_O_6_, obtained by the HRESIMS data (*m/z* 373.1652 [M + H]^+^, calcd for C_21_H_25_O_6_, 373.1651), is consistent with dihydroxylated derivative of the substrate **4**. The large coupling constant 15.6 Hz (*J*_Hα –Hβ_) indicated that the *E*-configuration at the α,β-double bond remained the same in the compound **10**. Compared to its substrate **4**, the ^1^H-NMR spectrum of **10** exhibited notable changes on a double bond in the prenyl group. Three new proton signals at δ_H_ 2.52 (1H, dd, *J* = 14.2, 10.4 Hz), 2.99 (1H, dd, *J* = 14.2, 1.7 Hz) and 3.61 (1H, dd, *J* = 10.4, 1.7 Hz) appeared in the ^1^H NMR spectrum. Similar changes were also observed in the ^13^C-NMR spectrum, including two oxygenated carbon signals at δ_C_ 78.5 and δ_C_ 72.5. This suggested that **10** was a dihydroxylated derivative of **4**. Locations of the two hydroxyl groups were confirmed to be at C-2″ and C-3″ by observation of HMBC correlations from H-4″/5″ (δ_H_ 1.26) to C-2″ (δ_C_ 78.5) and C-3″ (δ_C_ 72.5). On the basis of the ICD spectroscopic analysis, the absolute configuration at C-2″ was confirmed as *R* for the compound **10** (Appendix A). Therefore, structure of the compound **10** was elucidated as (2″*R*)-2”,3”-dihydroxylicochalcone H.

The molecular formula of the metabolite **11** was determined as C_21_H_22_O_5_ on the basis of the HRESIMS data (*m/z* 355.1544 [M + H]^+^, calculated for C_21_H_23_O_5_, 355.1545) which was identical to that of **7**. HPLC chromatography of the purified **11** showed two peaks at *t*_R_ of 17.15 and 19.56 min, respectively. UV λ_max_ of the two peaks were 267, 295 nm, and 276, 378 nm, respectively. Moreover, the ^1^H NMR pattern of **11** showed two sets of resonances in the spectra of ^1^H- and ^13^C-NMR. With the observations of two pairs of H-α and H-β proton signals at at δ_H_ 6.58 (1H, d, *J* = 12.8 Hz) and 7.00 (1H, d, *J* = 12.8 Hz), together with the signals at δ_H_ 7.70 (1H, d, *J* = 15.6 Hz) and 7.93 (1H, d, *J* = 15.6 Hz), it was revealed that **11** exists as a mixture of (*Z*)- and (*E*)-isomers. The integrals of (*Z*)-form and (*E*)-form (approximate ratio 0.3:1) indicated that the (*E*)-form was the major one. The NMR spectral data of the major *trans*-form showed three oxygenated proton signals at δ_H_ 3.67 (1H, dd, *J* = 7.5, 5.2 Hz, H-2″), 2.92 (1H, dd, *J* = 16.2, 5.2 Hz, H-1″a) and 2.60 (1H, dd, *J* = 16.2, 7.5 Hz, H-1″b) coupled with the signals at δ_c_ 68.4 (C-2″) and 30.6 (C-1″) on the basis of the combined analyses of ^1^H-NMR, ^13^C-NMR and HSQC of **11**, which indicated the epoxidation of the double bond between C-2″ and C-3″ of **4**. Further, location of the epoxidation was deduced to be between C-2″ and C-3″ based on the HMBC correlations from H-1″a and H-1″b to the carbon signals at δ_c_ 68.4 (C-2″) and 113.3 (C-5). Assignment of the minor *cis*-form was made possible by observing the same correlation patterns as those of *trans* isomer signals in the 1D and 2D NMR spectra. Therefore, the structure of **11** was confirmed as 2″,3″-expoxylicochalcone H (**11**).

### 2.2. Microbial Transformation of Licochalcones B *(**1**)*, C *(**2**)*, D *(**3**)* and H *(**4**)* by *M. hiemalis*

Microbial transformation of LB (**1**), LC (**2**) and LH (**4**) by M. hiemalis yielded the corresponding glucosylated metabolites **12**, **13** and **14**, respectively (Scheme 2).

Microbial transformation of licochalcone B (**1**) by M. hiemalis for five days produced the metabolite **12**. Compound **12** exhibited a molecular formula of C_22_H_24_O_10_ on the basis of its HRESIMS (*m/z* 449.1449 [M + H]^+^, calcd for C_22_H_25_O_10_, 449.1448), corresponding to the presence of an additional unit of C_6_H_10_O_5_ to **1**. The UV spectrum of **12** had λ_max_ at 208, 288 and 338 nm which is similar to that of **1**. The large coupling constant of 15.8 Hz (*J*_Hα–Hβ_) indicated that the *E*-configuration of the α,β-double bond remained the same in the compound **12**. The ^1^H- and ^13^C-NMR spectra of **12** showed characteristic signals of a sugar moiety. There were six carbon signals at δc 105.2, 77.7, 76.7, 74.4, 70.1, 61.2 and an anomeric proton signal at δ_H_ 4.74 (1H, d, *J* = 7.7 Hz). The other corresponding proton signals were assigned by the correlations in the spectrum of HSQC. Anomeric proton at δ_H_ 4.74 revealed that the glycoside linkage was formed between the anomeric hydroxyl group of the sugar moiety and the hydroxyl group of the C-3 (δ_H_ 138.7) of the aglycone based on their HMBC correlation. Acid hydrolysis of **12** afforded a d-glucose and the aglycone **1** which was identified by TLC and HPLC. All these evidences indicated that this sugar moiety was a β–d-glucopyranose. Therefore, the structure of **12** was assigned as licochalcone B 3-*O*-β-d-glucopyranoside.

Microbial transformation of licochalcone C (**2**) by *M. hiemalis* for three days produced the metabolite **13**. Compound **13** was isolated as a yellow amorphous powder, with a molecular formula of C_27_H_32_O_9_, which was determined by HRESIMS at *m/z* 501.2123 [M + H]^+^ (calcd for C_27_H_33_O_9_, 501.2125). In the comparison of its ^1^H- and ^13^C-NMR spectra with those of the substrate **2**, compound **13** displayed NMR resonance signals for the presence of a sugar moiety, showing an anomeric proton signal at δ_H_ 5.02 (1H, d, *J* = 7.2 Hz). Complete assignments of the ^1^H- and ^13^C-NMR signals of the sugar moiety were accomplished by HMBC and HSQC experiments. HMBC correlation from the anomeric proton signal at δ_H_ 5.02 (H-1’’’) to the carbon signal at δ_C_ 158.6 (C-4) confirmed the assignment of glycosylation at C-4. In the sugar analysis, d-glucopyranose was confirmed using TLC after acid hydrolysis of **13**. All these indicated the presence of a β-d-glucopyranose unit. The aglycone moiety of the ^1^H- and ^13^C-NMR spectra of the metabolite **13** displayed two sets of signals in the integral ratio of 1:0.2. Based on the observation of two pairs of H-α and H-β proton signals at δ_H_ 7.63 (1H, d, *J* = 15.7 Hz) and 7.93 (1H, d, *J* = 15.7 Hz), together with δ_H_ 6.58 (1H, d, *J* = 12.7 Hz) and 7.12 (1H, d, *J* = 12.7 Hz), **13** was clearly assigned as a mixture of (*E*)- and (*Z*)-isomers. On the basis of the above data and extensive 2D NMR experiments, structure of the compound **13** was assigned as licochalcone C 4-*O*-β-d-glucopyranoside.

Microbial transmformation of licochalcone H (**4**) by *M. hiemalis* for three days afforded metabolite **14**. Metabolite **14** was isolated as a yellow amorphous powder, with a molecular formula of C_27_H_32_O_9_, which was determined by HRESIMS at *m/z* 501.2122 [M + H]^+^ (calcd for C_27_H_33_O_9_, 501.2125). In the ^1^H-NMR spectrum of **14**, the presence of a sugar moiety was deduced by the appearance of an anomeric doublet proton at δ_H_ 5.00 with the coupling constant of 7.3 Hz, which indicated the β-orientation of the sugar moiety. Based on the information of a set of hexose moiety signals (δ_c_ 100.9, 77.2, 77.0, 73.6, 70.3 and 61.4) in the ^13^C-NMR spectrum, it was indicated that **14** was a glucosylated product of **4**. The identification was confirmed by acid hydrolysis and comparison with authentic glucose and the aglycone. The chemical shift of H-3 proton was significantly shifted downfield from δ_H_ 6.47 to 6.88, indicating that the glucosylation site was at 4-OH of the aglycone. Similarly to **13**, the ^1^H- and ^13^C-NMR supported the notion that the agalycone of **14** was an indivisible mixture of (*Z*)- and (*E*)-conformers. The aglycone part of the ^1^H- and ^13^C-NMR of the metabolite **14** displayed two sets of signals in the integral ratio of 0.4:1. On the basis of the observations of two pairs of H-α and H-β proton signals at δ_H_ 6.36 (1H, d, *J* = 12.7 Hz) and 7.11 (1H, d, *J* = 12.7 Hz), together with δ_H_ 7.63 (1H, d, *J* = 16.2 Hz) and 7.99 (1H, d, *J* = 16.2 Hz), **14** was clearly assignable as a mixture of (*Z*)- and (*E*)-isomers. Based on these analyses, the structure of **14** was elucidated as licochalcone H 4-*O*-β-d-glucopyranoside.

Generally, *trans*-chalcones are thermodynamically more stable than their corresponding *cis* isomers and most chalcones are thus isolated in *trans* form [28]. However, during the structure elucidation of the metabolites, the *trans*-*cis* isomerizations were observed by the analyses of the NMR spectra and HPLC chromatograms of metabolites **7**, **11**, **13,** and **14** (Scheme 1 and Scheme 2). The (*Z*)-form observed in the structures of these metabolites might be considered as artefacts formed during incubation, sample processing, or isolation procedure. There might be interconversion between the (*E*)-form and its corresponding (*Z*)-form because it was detected again after separation of the peaks of corresponding (*E*)- and (*Z*)-forms. Recently, two dihydrobenzofuran congeners of licochalcone A isoated from the roots of *Glycyrrhiza inflata* were found to rapidly isomerize and yield *trans* and *cis* isomers when solutions were exposed to sunlight [29]. Additionally, one metabolite of licochalcone A produced by human liver microsomes was observed as a mixture of *trans*- and *cis*-forms [30]. Based on these evidences, it could be deduced that some substituents introduced in the B ring of the retrochalcones may affect the stability of *trans*-form. Therefore, further studies will be necessary to evaluate the mechanism of *trans*-*cis* isomerization. Also the thermodynamics of retrochalcone isomerization should be considered in biological evaluation

## 3. Conclusions

Fungal transformation of the four licochalcones **1**–**4** by *A. niger* and *M. hiemalis* resulted in the formation of ten different metabolites **5**–**14**, including hydrogenated, dihydroxylated, expoxidized and glucosylated metabolites. Of them, the metabolites **5**, **6**, **8**, **9**, **10**, and **12** were confirmed to be in (*E*)-form; whereas the metabolites **7**, **11**, **13**, and **14** were confirmed as a mixture of (*E*)- and (*Z*)-isomers which were present in the integral ratios ranging from 1:0.3 to 1: 0.4 as evidenced by their ^1^H-NMR spectra. Although further investigations such as thermodynamic stability sutdies on these isomers may be needed, it was concluded that **7**, **11**, **13**, and **14** are present as interconverting mixtures of the major (*E*)- and minor (*Z*)-isomers.

Prenyl groups could affect the generation of the metabolites produced by *A. niger* and *M. hiemalis* (Figure 2). When prenyl groups were introduced in different positions of the retrochalcone backbone, it was observed that dihydroxylation or expoxidation of the corresponding substrates was preferentially performed on the prenyl side chains by *A. niger*. The position of glucosylation by *M. hiemalis* was preferred to be at the hydroxyl groups nearest to the prenyl groups. It was deduced that the flexibility of the prenyl substituents might have an effect on the regio-selectivity during the microbial transformation.

Microbial transformation is regarded as an efficient tool for the structural modification of bioactive natural and synthetic compounds [31]. Microbial transforamtion of the licochalcones can provide alternative approach to prepare the targeted licochaclcone derivatives under mild conditions by utilizing microbial system as biocatalysts.

## 4. Materials and Methods

### 4.1. General Experimental Procedures

Optical rotations were recorded with a 343 Plus polarimeter (Perkin Elmer, Waltham, MA, USA). UV spectra were recorded on a V-530 spectrophotometer (Jasco, Tokyo, Japan). IR spectra were obtained on a Jasco FT/IR 300-E spectrometer, and CD spectra were recorded on a Jasco J-815 CD spectrometer. NMR experiments were recorded using an Avance III 400 spectrometer (Bruker, Billerica, MA, USA) with TMS as the internal standard. HRESIMS were determined on Waters Synapt G2 QTOF (Waters, Milford, MA, USA). TLC was carried out on Merck silica gel F_254_-precoated glass plates and RP-18 F_254_s plates. Chromatography was performed on a Waters 1525 Binary HPLC pump connected to a 996 Photodiode Array (PDA) detector using Isco Allsphere ODS-2 (10 μm, 10 × 250 mm) and Nova-Pak C18 (4 μm, 3.9 × 150 mm) columns.

### 4.2. Preparation of Substrates

Licochalcones B (**1**) and D (**3**) were synthesized through acid-mediated Claisen-Schmidt condensation using 4-hydroxyacetophenone as a starting material [32]. And starting from 2,4-dihydroxybenzaldehyde, licochalcone C (**2**) and its regio-isomer licochalcone H (**4**) were synthesized by acid-mediated Claisen-Schmidt condensation [24]. The ^1^H-NMR data of licochlacones B (**1**), C (**2**), D (**3**), and H (**4**) agreed with data in the literature data [24,32]. The purity of each substrate was determined to be above 95% by HPLC analysis.

*Licochalcone B* (**1**): ^1^H NMR (DMSO-*d*_6_, 400 MHz, δ in ppm, *J* in Hz) δ 8.02 (2H, d, *J* = 8.7, H-2′,6′), 7.86 (1H, d, *J* = 15.9, H-β), 7.66 (1H, d, *J* = 15.9, H-α), 7.33 (1H, d, *J* = 8.7, H-6), 6.89 (2H, d, *J* = 8.7, H-3′,5′), 6.64 (1H, d, *J* = 8.7, H-5), 3.78 (3H, s, 2-OMe).

*Licochalcone C* (**2**): ^1^H NMR (CDCl_3_, 400 MHz, δ in ppm, *J* in Hz) δ 8.02 (1H, d, *J* = 15.7, H-β), 7.99 (2H, d, *J* = 8.8, H-2′,6′), 7.51 (1H, d, *J* = 15.7, H-α), 7.47 (1H, d, *J* = 8.8, H-6), 6.96 (2H, d, *J* = 8.8, H-3′,5′), 6.70 (1H, d, *J* = 8.8, H-5), 5.23 (1H, m, H-2″), 3.74 (3H, s, 2-OMe), 3.45 (2H, d, *J* = 6.9, H-1″), 1.82 (3H, s, H-4″), 1.74 (3H, s, H-5″).

*Licochalcone D* (**3**): ^1^H NMR (CD_3_OD, 400 MHz, δ in ppm, *J* in Hz) δ 7.92 (1H, d, *J* = 15.7, H-β), 7.82 (1H, d, *J* = 8.7, H-6′), 7.80 (1H, s, H-2′), 7.60 (1H, d, *J* = 15.7, H-α), 7.18 (1H, d, *J* = 8.6, H-6), 6.85 (1H, d, *J* = 8.7, H-5′), 6.64 (1H, d, *J* = 8.6, H-5), 5.36 (1H, m, H-2″), 3.85 (3H, s, 2-OMe), 3.34 (2H, d, *J* = 7.4, H-1″), 1.76 (3H, s, H-4″), 1.74 (3H, s, H-5″).

*Licochalcone H* (**4**): ^1^H NMR (CD_3_OD, 400 MHz, δ in ppm, *J* in Hz) δ 7.99 (1H, d, *J* = 15.6, H-β), 7.95 (2H, d, *J* = 8.8, H-2′,6′), 7.55 (1H, d, *J* = 15.6, H-α), 7.39 (1H, s, H-6), 6.89 (2H, d, *J* = 8.8, H-3′,5′), 6.47 (1H, s, H-3), 5.32 (1H, m, H-2″), 3.87 (3H, s, 2-OMe), 3.26 (2H, d, *J* = 7.3, H-1″), 1.75 (3H, s, H-4″), 1.74 (3H, s, H-5″).

### 4.3. Microorganisms and Culture Media

All the microorganisms were obtained from the Korean Collection for Type Cultures (KCTC, Jeongeup, Korea) and Korean Culture Center of Microorganisms (KCCM, Seoul, Korea). Eight cultures were used for the preliminary screening process as listed below: *Absidia coerulea* KCTC 6936, *Aspergillus fumigatus* 6145, *Cunninghamella elegans* var. *elegans* 6992, *Mortierella ramanniana* var. *angulispora* 6137*, Mucor hiemalis* 26779, *Penicillium chrysogenum* 6933, *Trichoderma koningii* 6042, and *Aspergillus niger* KCCM 60332.

All the ingredients for microbial media, including dextrose, peptone, malt extract, yeast extract, and potato dextrose broth were purchased from Becton, Dickinson and Co. (Sparks, MD, USA). Two types of media were used in the fermentation experiments and are listed below: *A. coerulea*, *A. fumigatus*, *A. niger*, *M. hiemalis*, *P. chrysogenum*, and *T. koningii* were cultured on malt medium (malt extract 20 g/L, dextrose 20 g/L, peptone 1 g/L); *C. elegans* var. *elegans* and *M. ramanniana* var. *angulispora* were cultured on potato dextrose medium (24 g/L).

### 4.4. Procedure for Microbial Transformation

Microbial metabolism studies were carried out according to the standard two-stage procedure [21]. Briefly, the actively growing microbial cultures were inoculated in 250 mL Erlenmeyer flasks containing 50 mL of a suitable medium, and incubated with gentle agitation (200 rpm) at 25 °C in a temperature-controlled shaking incubator. The DMSO solutions (20 mg/mL, 100 μL) of the substrates (**1**, **2**, **3** or **4**) were added to each flask 24 h after inoculation, and further incubated at the same conditions for another 5 days. Sampling and TLC monitoring were performed at an interval of 24 h. UV (254 and 365 nm) and anisaldehyde-sulfuric acid reagent were used for identification of metabolites. Culture controls were carried out as a result of enzymatic activity, but not a consequence of degradation or non-metabolic changes.

Similarly, the preparative-scale fermentations were carried out in 500 mL flasks containing 150 mL of culture medium, and the selected microorganisms were pre-cultured under the culture conditions mentioned above for 24 h to obtain sufficient amounts for biotransformation. After that, 50 mg of **1**, 80 mg of **2**, 35 mg of **3** or 80 mg of **4** dissolved in DMSO (20 mg/mL) were used for preparative-scale fermentations. The other procedures and culture conditions were same as those of the screening experiments.

### 4.5. Extraction and Isolation of Metabolites

After incubation, the cultures of **1**, **2**, **3**, or **4** were extracted with equal volume of EtOAc two times, and the organic layer was collected and concentrated. The EtOAc extract of **1** incubated with *A. niger* was subjected to reversed-phased HPLC with a gradient solvent system of 35% MeOH to 45% MeOH to afford **5** (5.8 mg, t_R_ = 10.92 min) at a flow rate of 1.0 mL/min. The EtOAc extract of **1** incubated with *M. hiemalis* was subjected to reversed-phased HPLC with an isocratic solvent system of 34% MeOH to afford **12** (4.0 mg, t_R_ = 8.79 min) at a flow rate of 1.0 mL/min. The EtOAc extract of **2** incubated with *A. niger* was subjected to reversed-phased HPLC with a gradient solvent system of 53% MeOH to 69% MeOH to afford **6** (4.6 mg, t_R_ = 22.17 min) and **7** (5.2 mg, t_R_ = 30.46 min) at a flow rate of 2.0 mL/min. The EtOAc extract of **2** incubated with *M. hiemalis* was subjected to reversed-phased HPLC with a gradient solvent system of 65% MeOH to 80% MeOH to afford **13** (5.0 mg, t_R_ = 12.94 min) at a flow rate of 2.0 mL/min. The EtOAc extract of **3** with incubated with *A. niger* was subjected to reversed-phased HPLC with a gradient solvent system of 58% MeOH to 65% MeOH to afford **8** (2.5 mg, t_R_ = 12.62 min) and **9** (3.2 mg, t_R_ = 20.08 min) at a flow rate of 2.0 mL/min. The EtOAc extract of **4** incubated with *A. niger* was subjected to reversed-phased HPLC with a gradient solvent system of 60% MeOH to 70% MeOH to afford **10** (4.3 mg, t_R_ = 14.40 min) and **11** (5.5 mg, t_R_ = 20.48 min) at a flow rate of 2.0 mL/min. The EtOAc extract of **4** incubated with *M. hiemalis* was subjected to reversed-phased HPLC with a gradient solvent system of 60% MeOH to 75% MeOH to afford **14** (5.0 mg, t_R_ = 16.51 min) at a flow rate of 2.0 mL/min.

### 4.6. Spectroscopic Data of Metabolites

*α,β-Dihydrolicochalcone B* (**5**): yellow powder; UV λ_max_ (MeOH): 208, 276 nm; IR ν_max_: 3599, 3178, 1648, 1276, 1069, 798 cm^−1^; HRESIMS *m/z*: 311.0896 [M + Na]^+^ (calcd for C_16_H_16_O_5_Na, 311.0895); ^1^H-NMR (CD_3_OD, 400 MHz, δ in ppm, *J* in Hz) δ 7.89 (2H, d, *J* = 8.7, H-2′,6′), 6.84 (2H, d, *J* = 8.7, H-3′,5′), 6.52 (1H, d, *J* = 8.3, H-6) 6.49 (1H, d, *J* = 8.3, H-5), 3.80 (3H, s, 2-OMe), 3.15 (2H, t, *J* = 7.4, H-α), 2.87 (2H, t, *J* = 7.4, H-β); ^13^C-NMR (CD_3_OD, 100 MHz): 200.0 (C=O), 162.3 (C-4′), 146.2 (C-2), 144.7 (C-4), 138.1 (C-3), 130.5 (C-2′,6′), 128.6 (C-1′), 125.2 (C-1), 119.1 (C-6), 114.9 (C-3′,5′), 110.5 (C-5), 59.5 (2-OMe), 39.2 (C-α), 25.1 (C-β).

*(2″R)-2”,3”-Dihydroxylicochalcone C* (**6**): yellow powder; ${\left[\alpha \right]}_{D}^{20}$ +3.6° (*c* 0.10, MeOH); UV λ_max_ (MeOH): 207, 352 nm; IR ν_max_: 3595, 3391, 1642, 1282, 1073, 808 cm^−1^; HRESIMS *m/z*: 395.1472 [M + Na]^+^ (calcd for C_21_H_24_O_6_Na, 395.1471); ^1^H-NMR (CD_3_OD, 400 MHz, δ in ppm, *J* in Hz) δ 8.00 (1H, d, *J* = 15.6, H-β), 7.99 (2H, d, *J* = 8.8, H-2′,6′), 7.65 (1H, d, *J* = 8.6, H-6), 7.63 (1H, d, *J* = 15.6, H-α), 6.90 (2H, d, *J* = 8.8, H-3′,5′), 6.72 (1H, d, *J* = 8.6, H-5), 3.80 (3H, s, 2-OMe), 3.65 (1H, dd, *J* = 10.3, 2.0, H-2″), 3.12 (1H, dd, *J* = 14.2, 2.0, H-1″a), 2.69 (1H, dd, *J* = 14.2, 10.3, H-1″b), 1.27 (6H, s, H-4″,5″); ^13^C-NMR (CD_3_OD, 100 MHz, δ in ppm) 189.9 (C=O), 162.4 (C-4′), 160.1 (C-4), 160.1 (C-2), 139.6 (C-β), 130.8 (C-2′,6′), 129.8 (C-1′), 126.9 (C-α), 120.9 (C-1), 119.9(C-6), 119.1 (C-3), 115.0 (C-3′,5′), 112.5 (C-5), 78.9 (C-2″), 72.6 (C-3″), 61.9 (2-OMe), 25.9 (C-1″), 24.1 (C-5″), 23.8 (C-4″).

*2”,3”-Epoxylicochalcone C* (**7**, a mixture of *trans* and *cis* forms): yellow powder; ${\left[\alpha \right]}_{D}^{20}$ +7.1° (*c* 0.20, MeOH); UV λ_max_ (MeOH): 208, 354 nm; IR ν_max_: 3629, 3200, 1595, 1220, 1071, 814 cm^−1^; HRESIMS *m/z*: 377.1376 [M + Na]^+^ (calcd for C_21_H_22_O_5_Na, 377.1365).

*Trans* form of **7**: ^1^H-NMR (CD_3_OD, 400 MHz, δ in ppm, *J* in Hz) δ 8.01 (1H, d, *J* = 15.6, H-β), 7.99 (2H, d, *J* = 8.8, H-2′,6′), 7.66 (1H, d, *J* = 8.6, H-6), 7.64 (1H, d, *J* = 15.6, H-α), 6.90 (2H, d, *J* = 8.8, H-3′,5′), 6.65 (1H, d, *J* = 8.6, H-5), 3.81 (3H, s, 2-OMe), 3.80 (1H, dd, *J* = 7.0, 5.0, H-2″), 3.04 (1H, dd, *J* = 17.1, 5.0, H-1″a), 2.73 (1H, dd, *J* = 17.1, 7.0, H-1″b), 1.35 (3H, s, H-4″), 1.30 (3H, s, H-5″); ^13^C-NMR (CD_3_OD, 100 MHz, δ in ppm) 189.8 (C=O), 162.4 (C-4′), 159.4 (C-2), 156.9 (C-4), 138.9 (C-β), 130.9 (C-2′,6′), 129.8 (C-1′), 126.5 (C-α), 120.0 (C-1), 119.3(C-6), 115.0 (C-3′,5′), 114.2 (C-3), 113.6 (C-5), 77.3 (C-3″), 68.3 (C-2″), 60.8 (2-OMe), 25.8 (C-1″), 24.4 (C-4″), 20.0 (C-5″).

*Cis* form of **7**: ^1^H-NMR (CD_3_OD, 400 MHz, δ in ppm, *J* in Hz) δ 7.82 (2H, d, *J* = 8.8, H-2′,6′), 7.11 (1H, d, *J* = 12.8, H-β), 7.06 (1H, d, *J* = 8.7, H-6), 6.76 (2H, d, *J* = 8.8, H-3′,5′), 6.55 (1H, d, *J* = 12.8, H-α), 6.36 (1H, d, *J* = 8.7, H-5), 3.73 (3H, s, 2-OMe), 3.70 (1H, dd, *J* = 7.2, 5.3, H-2″), 2.92 (1H, dd, *J* = 17.3, 5.5, H-1″a), 2.59 (1H, dd, *J* = 17.3, 7.2, H-1″b), 1.28 (3H, s, H-4″), 1.21 (3H, s, H-5″); ^13^C-NMR (CD_3_OD, 100 MHz, δ in ppm) 194.7 (C=O), 162.5 (C-4′), 157.5 (C-2), 154.9 (C-4), 134.1 (C-β), 131.3 (C-2′,6′), 129.0 (C-1′), 128.6 (C-6), 125.1 (C-α), 120.6 (C-1), 114.8 (C-3′,5′), 113.6 (C-3), 112.2 (C-5), 76.8 (C-3″), 68.5 (C-2″), 60.1 (2-OMe), 25.8 (C-1″), 24.3 (C-4″), 19.7 (C-5″).

*(2″R)-2”,3”-Dihydroxylicochalcone D* (**8**): yellow powder; ${\left[\alpha \right]}_{D}^{20}$ +5.1° (*c* 0.15, MeOH); UV λ_max_ (MeOH): 209, 363 nm; IR ν_max_: 3627, 2924, 1591, 1264, 1063, 795 cm^−1^; HRESIMS *m/z*: 389.1603 [M + H]^+^ (calcd for C_21_H_25_O_7_, 389.1600); ^1^H-NMR (CD_3_OD, 400 MHz, δ in ppm, *J* in Hz) δ 8.00 (1H, d, *J* = 15.7, H-β), 7.97 (1H, d, *J* = 1.7, H-2′), 7.87 (1H, dd, *J* = 8.1, 1.7, H-6′), 7.67 (1H, d, *J* = 15.7, H-α), 7.26 (1H, d, *J* = 8.3, H-6), 6.90 (1H, d, *J* = 8.1, H-5′), 6.66 (1H, d, *J* = 8.3, H-5), 3.85 (3H, s, 2-OMe), 3.66 (1H, dd, *J* = 10.4, 1.3, H-2″), 3.13 (1H, dd, *J* = 13.8, 1.3, H-1″a), 2.60 (1H, dd, *J* = 13.8, 10.4, H-1″b), 1.26 (6H, s, H-4″,5″); ^13^C-NMR (CD_3_OD, 100 MHz, δ in ppm) 190.2 (C=O), 160.7 (C-4′), 149.4 (C-4), 148.5 (C-2), 139.4 (C-β), 138.3 (C-3), 132.5 (C-2′), 129.8 (C-1′), 128.7 (C-6′), 126.9 (C-3′), 120.1 (C-1), 119.2 (C-α), 118.7 (C-6), 114.8 (C-5′), 111.3 (C-5), 78.1 (C-2″), 72.5 (C-3″), 60.4 (2-OMe), 32.5 (C-1″), 24.3 (C-5″), 23.7 (C-4″).

*2”,3”-Epoxylicochalcone D* (**9**): yellow powder; ${\left[\alpha \right]}_{D}^{20}$ −2.4° (*c* 0.15, MeOH); UV λ_max_ (MeOH): 209, 363 nm; IR ν_max_: 3625, 2924, 1575, 1258, 1059, 770 cm^−1^; HRESIMS *m/z*: 371.1497 [M + H]^+^ (calcd for C_21_H_23_O_6_, 371.1495); ^1^H-NMR (CD_3_OD, 400 MHz, δ in ppm, *J* in Hz) δ 7.89 (1H, d, *J* = 15.6, H-β), 7.87 (1H, d, *J* = 2.2, H-2′), 7.84 (1H, dd, *J* = 8.3, 2.2, H-6′), 7.64 (1H, d, *J* = 15.6, H-α), 7.24 (1H, d, *J* = 8.6, H-6), 6.87 (1H, d, *J* = 8.3, H-5′), 6.66 (1H, d, *J* = 8.6, H-5), 3.85 (3H, s, 2-OMe), 3.82 (1H, dd, *J* = 7.0, 5.0, H-2″), 3.11 (1H, dd, *J* = 16.8, 5.0, H-1″a), 2.82 (1H, dd, *J* = 16.8, 7.0, H-1″b), 1.36 (3H, s, H-4″), 1.31 (3H, s, H-5″); ^13^C-NMR (CD_3_OD, 100 MHz, δ in ppm) 190.0 (C=O), 157.8 (C-4′), 149.5 (C-4), 148.6 (C-2), 139.8 (C-β), 138.3 (C-3), 131.1 (C-2′), 130.6 (C-1′), 128.3 (C-6′), 120.1 (C-3′), 120.0 (C-1), 119.1 (C-α), 118.9 (C-6), 116.8 (C-5′), 111.4 (C-5), 78.0 (C-3″), 68.6 (C-2″), 60.4 (2-OMe), 30.6 (C-1″), 24.5 (C-4″), 20.2 (C-5″),.

*(2″R)-2”,3”-Dihydroxylicochalcone H* (**10**): yellow powder; ${\left[\alpha \right]}_{D}^{20}$ +2.2° (*c* 0.20, MeOH); UV λ_max_ (MeOH): 208, 379 nm; IR ν_max_: 3422, 2930, 1647, 1290, 1063, 836 cm^−1^; HRESIMS *m/z*: 373.1652 [M + H]^+^ (calcd for C_21_H_25_O_6_, 373.1651); ^1^H-NMR (CD_3_OD, 400 MHz, δ in ppm, *J* in Hz) δ 8.06 (1H, d, *J* = 15.6, H-β), 7.99 (2H, d, *J* = 8.8, H-2′,6′), 7.63 (1H, d, *J* = 15.6, H-α), 7.58 (1H, s, H-6), 6.90 (2H, d, *J* = 8.8, H-3′,5′), 6.49 (1H, s, H-3), 3.87 (3H, s, 2-OMe), 3.61 (1H, dd, *J* = 10.4, 1.7, H-2″), 2.99 (1H, dd, *J* = 14.2, 1.7, H-1″a), 2.52 (1H, dd, *J* = 14.2, 10.4, H-1″b), 1.26 (6H, s, H-4″,5″); ^13^C-NMR (CD_3_OD, 100 MHz, δ in ppm) 190.2 (C=O), 162.2 (C-4′), 159.9 (C-4), 159.3 (C-2), 140.0 (C-β), 131.6 (C-6), 130.7 (C-2′,6′), 130.1 (C-1′), 119.4 (C-5), 117.8 (C-α), 115.1 (C-1), 114.9 (C-3′,5′), 98.6 (C-3), 78.5 (C-2″), 72.5 (C-3″), 54.8 (2-OMe), 31.8 (C-1″), 24.3 (C-4″), 23.7 (C-5″).

*2”,3”-Epoxylicochalcone H* (**11**): yellow powder; ${\left[\alpha \right]}_{D}^{20}$ +1.5° (*c* 0.14, MeOH); UV λ_max_ (MeOH): 208, 292, 371 nm; IR ν_max_: 3680, 2926, 1593, 1284, 1057, 840 cm^−1^; HRESIMS *m/z*: 355.1544 [M + H]^+^ (calcd for C_21_H_23_O_5_, 355.1545);

*Trans* form of **11**: ^1^H-NMR (DMSO-*d*_6_, 400 MHz, δ in ppm, *J* in Hz) δ 8.03 (2H, d, *J* = 8.6, H-2′,6′), 7.93 (1H, d, *J* = 15.6, H-β), 7.72 (1H, s, H-6), 7.70 (1H, d, *J* = 15.6, H-α), 6.90 (2H, d, *J* = 8.6, H-3′,5′), 6.44 (1H, s, H-3), 3.82 (3H, s, 2-OMe), 3.67 (1H, dd, *J* = 7.5, 5.2, H-2″), 2.92 (1H, dd, *J* = 16.2, 5.2, H-1″a), 2.60 (1H, dd, *J* = 16.2, 7.5, H-1″b), 1.30 (3H, s, H-4″), 1.21 (3H, s, H-5″); ^13^C-NMR (DMSO-d_6_, 100 MHz, δ in ppm) 187.6 (C=O), 162.3 (C-4′), 158.6 (C-2), 156.9 (C-4), 138.0 (C-β), 131.3 (C-2′,6′), 130.3 (C-6), 130.0 (C-1′), 119.0 (C-α), 116.1 (C-1), 115.8 (C-3′,5′), 113.3 (C-5), 100.8 (C-3), 78.5 (C-3″), 68.4 (C-2″), 56.2 (2-OMe), 30.6 (C-1″), 26.1 (C-4″), 21.3 (C-5″);

*Cis* form of **11**: ^1^H-NMR (DMSO-*d*_6_, 400 MHz, δ in ppm, *J* in Hz) δ 7.77 (2H, d, *J* = 8.6, H-2′,6′), 7.14 (1H, s, H-6), 7.00 (1H, d, *J* = 12.8, H-β), 6.81 (2H, d, *J* = 8.6, H-3′,5′), 6.58 (1H, d, *J* = 12.8, H-α), 6.30 (1H, s, H-3), 3.65 (3H, s, 2-OMe), 3.56 (1H, dd, *J* = 7.2, 5.2, H-2″), 2.67 (1H, dd, *J* = 16.0, 5.2, H-1″a), 2.39 (1H, dd, *J* = 16.0, 7.7, H-1″b), 1.24 (3H, s, H-4″), 1.12 (3H, s, H-5″); ^13^C-NMR (DMSO-d_6_, 100 MHz, δ in ppm) 192.3 (C=O), 162.3 (C-4′), 157.1 (C-2), 155.1 (C-4), 133.9 (C-β), 131.9 (C-6), 131.4 (C-2′,6′), 129.5 (C-1′), 124.8 (C-α), 116.9 (C-1), 115.6 (C-3′,5′), 111.7 (C-5), 99.5 (C-3), 78.0 (C-3″), 68.4 (C-2″), 55.8 (2-OMe), 30.6 (C-1″), 26.0 (C-4″), 20.9 (C-5″).

*Licochalcone B 3-O-β-**d-glucopyranoside* (**12**): yellow powder; UV λ_max_ (MeOH): 208, 288, 338 nm; IR ν_max_: 3568, 3448, 1641, 1000, 786 cm^−1^; HRESIMS *m/z*: 449.1449 [M + H]^+^ (calcd for C_22_H_25_O_10_, 449.1448); ^1^H- NMR (DMSO-*d*_6_, 400 MHz, δ in ppm, *J* in Hz) δ 8.00 (2H, d, *J* = 8.8, H-2′,6′), 7.82 (1H, d, *J* = 15.8, H-β), 7.66 (1H, d, *J* = 15.8, H-α), 7.63 (1H, d, *J* = 8.8, H-6), 6.88 (2H, d, *J* = 8.8, H-3′,5′), 6.66 (1H, d, *J* = 8.8, H-5), 4.74 (1H, d, *J* = 7.7, H-1″), 3.89 (3H, s, 2-OMe), 3.65 (1H, dd, *J* = 11.7, 1.7, H-6″a), 3.49 (1H, m, H-6″b), 3.34 (1H, m, H-2″), 3.26 (1H, m, H-5″), 3.20 (1H, m, H-4″), 3.19 (1H, m, H-3″). ^13^C-NMR (CD3OD, 400 MHz): 187.7 (C=O), 162.4 (C-4′), 153.9 (C-2), 153.9 (C-4), 138.7 (C-3), 138.1 (C-β), 131.4 (C-2′,6′), 129.9 (C-1′), 124.6 (C-6), 119.9 (C-α), 119.8 (C-1), 115.8 (C-3′,5′), 113.4 (C-5), 105.2 (C-1″), 77.7 (C-3″), 76.7 (C-5″), 74.4 (C-2″), 70.1 (C-4″), 62.6 (2-OMe), 61.2 (C-6″).

*Licochalcone C 4-O-β-**d-glucopyranoside* (**13**): yellow powder; UV λ_max_ (MeOH): 208, 343 nm; IR ν_max_: 3655, 3328, 1590, 1224, 1072, 838 cm^−1^; HRESIMS *m/z*: 501.2123 [M + H]^+^ (calcd for C_27_H_33_O_9_, 501.2125).

*Trans* form of **13**: ^1^H-NMR (CD_3_OD, 400 MHz, δ in ppm, *J* in Hz) δ 8.02 (2H, d, *J* = 8.8, H-2′,6′), 7.93 (1H, d, *J* = 15.7, H-β), 7.72 (1H, d, *J* = 8.8, H-6), 7.63 (1H, d, *J* = 15.7, H-α), 6.91 (2H, d, *J* = 8.8, H-3′,5′), 7.05 (1H, d, *J* = 8.8, H-5), 5.26 (1H, m, H-2″), 5.02 (1H, d, *J* = 7.2, H-1’’’), 3.91 (1H, dd, *J* = 11.9, 1.7, H-6’’’a), 3.75 (3H, s, 2-OMe), 3.72 (1H, m, H-6’’’b), 3.52-3.46 (1H, H-2’’’), 3.50-3.44 (2H, m, H-3’’’,5’’’), 3.49-3.42 (1H, m, H-4’’’), 3.45 (2H, overlapped, H-1″), 1.80 (3H, s, H-4″), 1.67 (3H, s, H-5″); ^13^C-NMR (CD_3_OD, 100 MHz, δ in ppm) 189.9 (C=O), 162.5 (C-4′), 159.0 (C-2), 158.6 (C-4), 139.0 (C-β), 131.0 (C-2′,6′), 130.9 (C-3″), 129.6 (C-1′), 126.4 (C-6), 125.0 (C-3), 122.8 (C-2″), 122.4 (C-1), 120.5 (C-α), 115.2 (C-3′,5′), 111.1 (C-5), 100.5 (C-1’’’), 76.8 (C-3’’’), 76.6 (C-5’’’), 73.6 (C-2’’’), 69.9 (C-4’’’), 61.8 (2-OMe), 61.1 (C-6’’’), 24.6 (C-5″), 22.6 (C-1″), 16.8 (C-4″);

*Cis* form of **13**: ^1^H-NMR (CD_3_OD, 400 MHz, δ in ppm, *J* in Hz) δ 7.83 (2H, d, *J* = 8.8, H-2′,6′), 7.12 (1H, d, *J* = 12.7, H-β), 7.10 (1H, d, *J* = 8.8, H-6), 6.78 (2H, d, *J* = 8.8, H-3′,5′), 6.77 (1H, d, *J* = 8.8, H-5), 6.58 (1H, d, *J* = 12.7, H-α), 5.14 (1H, m, H-2″), 4.86 (1H, overlapped, H-1’’’), 3.85 (1H, m, H-6’’’a), 3.72 (1H, m, H-6’’’b), 3.70 (3H, s, 2-OMe), 3.52-3.46 (1H, H-2’’’), 3.50-3.44 (2H, m, H-3’’’,5’’’), 3.49-3.42 (1H, m, H-4’’’), 3.48 (2H, overlapped, H-1″), 1.75 (3H, s, H-4″), 1.64 (3H, s, H-5″); ^13^C-NMR (CD_3_OD, 100 MHz, δ in ppm) 194.9 (C=O), 162.6 (C-4′), 157.2 (C-2), 156.9 (C-4), 133.8 (C-β), 131.5 (C-2′,6′), 130.5 (C-3″), 128.8 (C-1′), 128.2 (C-6), 126.1 (C-α), 124.4 (C-3), 123.3 (C-1), 123.0 (C-2″), 114.9 (C-3′,5′), 110.1 (C-5), 100.7 (C-1’’’), 76.9 (C-3’’’), 76.6 (C-5’’’), 73.6 (C-2’’’), 69.9 (C-4’’’), 61.1 (C-6’’’), 60.9 (2-OMe), 24.6 (C-5″), 22.5 (C-1″), 16.8 (C-4″).

*Licochalcone H 4-O-β-**d-glucopyranoside* (**14**): yellow powder; UV λ_max_ (MeOH): 207, 287, 358 nm; IR ν_max_: 3723, 3160, 1602, 1227, 1064, 854 cm^−1^; HRESIMS *m/z*: 501.2122 [M + H]^+^ (calcd for C_27_H_33_O_9_, 501.2125).

*Trans* form of **14**: ^1^H-NMR (CD_3_OD, 400 MHz, δ in ppm, *J* in Hz) δ 7.99 (1H, d, *J* = 16.2, H-β), 7.95 (2H, d, *J* = 8.8, H-2′,6′), 7.63 (1H, d, *J* = 16.2, H-α), 7.45 (1H, s, H-6), 6.90 (2H, d, *J* = 8.8, H-3′,5′), 6.88 (1H, s, H-3), 5.34 (1H, m, H-2″), 5.00 (1H, d, *J* = 7.3, H-1’’’), 3.92 (1H, overlapped, H-6’’’a), 3.92 (3H, s, 2-OMe), 3.66 (1H, m, H-6’’’b), 3.52-3.46 (1H, m, H-2’’’), 3.49-3.35 (1H, m, H-3’’’, 5’’’), 3.35-3.33 (1H, m, H-4’’’), 3.34 (2H, overlapped, H-1″), 1.74 (6H, s, H-4″, 5″); ^13^C-NMR (CD_3_OD, 100 MHz, δ in ppm) 190.2 (C=O), 162.3 (C-4′), 158.7 (C-2), 158.6 (C-4), 139.6 (C-β), 131.8 (C-3″), 130.8 (C-2′,6′), 129.9 (C-1′), 129.6 (C-6), 123.2 (C-5), 122.7 (C-2″), 119.5 (C-α), 117.3 (C-1), 115.0 (C-3′,5′), 100.9 (C-1’’’), 99.0 (C-3), 77.2 (C-3’’’), 77.0 (C-5’’’), 73.6 (C-2’’’), 70.3 (C-4’’’), 61.4 (C-6’’’), 55.0 (2-OMe), 27.3 (C-1″), 24.6 (C-4″), 16.6 (C-5″);

*Cis* form of **14**: ^1^H-NMR (CD_3_OD, 400 MHz, δ in ppm, *J* in Hz) δ 7.80 (2H, d, *J* = 8.7, H-2′,6′), 7.11 (1H, d, *J* = 12.7, H-β), 6.88 (1H, s, H-6), 6.76 (1H, s, H-3), 6.73 (2H, d, *J* = 8.7, H-3′,5′), 6.36 (1H, d, *J* = 12.7, H-α), 5.40 (1H, m, H-2″), 4.85 (1H, overlapped, H-1’’’), 3.92 (1H, overlapped, H-6’’’a), 3.76 (3H, s, 2-OMe), 3.66 (1H, m, H-6’’’b), 3.52-3.46 (1H, m, H-2’’’), 3.49-3.35 (1H, m, H-3’’’, 5’’’), 3.35-3.33 (1H, m, H-4’’’), 3.13 (2H, overlapped, H-1″), 1.63 (3H, s, H-4″), 1.55 (3H, s, H-5″); ^13^C-NMR (CD_3_OD, 100 MHz, δ in ppm) 196.0 (C=O), 162.5 (C-4′), 156.7 (C-4), 156.4 (C-2), 133.0 (C-β), 131.9 (C-3″), 131.6 (C-2′,6′), 130.6 (C-6), 128.6 (C-1′), 124.9 (C-α), 122.2 (C-2″), 122.0 (C-5), 118.2 (C-1), 114.8 (C-3′,5′), 101.2 (C-1’’’), 98.8 (C-3), 77.1 (C-3’’’), 76.9 (C-5’’’), 73.6 (C-2’’’), 70.2 (C-4’’’), 61.3 (C-6’’’), 54.7 (2-OMe), 26.6 (C-1″), 24.5 (C-4″), 16.4 (C-5″).

### 4.7. Acid Hydrolysis

Each solution of metabolites **12**–**14** (each 1 mg) in 2N HCl was heated for 2 h. After cooling, the reaction mixture was neutralized and partitioned with EtOAc. The organic and aqueous extracts were analyzed by HPLC and TLC, respectively. The monosaccharide of each metabolite was confirmed to be d-glucose by comparing its R_f_ value with that of authentic d-glucose on TLC plate and the aglycone of each metabolite was confirmed by comparing the retention times with those of aglycones (**1**, **2**, **4**) on HPLC.

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
