# Peer review of "Microbial Transformation of Licochalcones"

_molecules, 2019, doi:10.3390/molecules25010060_

Round 1

Reviewer 1 Report

This manuscript written by Yina Xiao and coworkers described the microbial transformation of licochalcones using Aspergillus niger and Mucor hiemalis. The authors spent a lot of efforts to determine the structure of 10 new derivatives.  I think the structural elucidation part of those new compounds is correct. It is noted that the microbial transformation of the chosen two fungi were studied before (ref 16). Which means the novelty of this study is not very high. I think this paper can be accepted as present form.

Author Response

We would like to express our appreciation for your attention and valuable comments on this study. Thanks for spending time to read this paper and finding it interesting.

Some mistakes were found in the manuscript and were corrected in green, and ‘Abstract’ has been revised for clarification of wording. Revisions have also been made in the ‘Introduction’ (Page 1: lines 33-40) and in the ‘Conclusions’ (Page 7: lines 240-245 and 257-260) in red.

Reviewer 2 Report

Recommendation: Accept after minor revision

In the present manuscript authors describes the importance of licochalcones in medicinal chemistry. Authors have done an excellent research work and very useful for synthetic and medicinal chemistry community. On the other hand, the authors did not collect sufficient literature, some of the recent literature about research against chalcones and biological importance of chalcones were missing. The manuscript needs extensive revisions.

Minor Revision Comments:

Please carefully select the keywords. In introduction part, please cite the following missing recent articles:

Bioorganic Chemistry 80 (2018) 86-93; Bioorganic Chemistry 81 (2018) 389-395; European Journal of Medicinal Chemistry 162 (2019) 364-377; Bioorganic Chemistry 91 (2019) 103133.

The conclusion part is not attractive to a broad readership of medicinal chemistry community, please rewrite that part. The manuscript needs extensive improvement by an native English language speaker.

Author Response

These are very helpful comments to raise the manuscript’s quality. Revision has been made according to the reviewer’s suggestions.

(1) The keywords, “licochalcone; retrochalcone; fungal transformation; trans-cis isomerization”, have been revised to “licorice; licochalcone; retrochalcone; fungal transformation” (Page 1: line 20).

(2) We are thankful to the reviewer’s efforts to provide some of the recent literatures about research against chalcones and their biological importance. After carefully reading the suggested qualified references, two of them [Bioorg. Chem. 80 (2018) 86-93; Bioorg. Chem. 91 (2019) 103133] were found to be more relevant to our present study and were therefore cited as Ref. [8] and [19] in the introduction part (Page 1: lines 33-40, in red). Additionally, two more recent literatures about biological importance of chalcones have been added in the ‘References’ as Ref. [17] and [18] (in red) to provide more sufficient literature.

(3) Following the reviewer’s comments, the conclusion part has been rewritten which could be more attractive to a broad readership of medicinal chemistry community (Page 7: lines 240-245 and 257-260, in red). The manuscript has been revised and improved by a native English speaker.

In addition, some mistakes were found in the manuscript and were corrected in green, and ‘Abstracts’ has been revised for clarification of wording.

Reviewer 3 Report

This Paper can be accepted after modification.

1. In conclusion the author should mention that the ten different metabolites they found whether those are trans-cis isomer or trans isomer only in a sentence.
2. The thermodynamic stability study of the metabolites could be performed which will add more value to this paper however author has mentioned to performed it in future study.
3. Author has performed several characterisation methods and most importantly NMR studies which has confirmed the elucidation of different metabolites.

Author Response

(1) Firstly, we would like to thank the reviewer for taking time to review and forward valuable comments to enrich the content of the manuscript. In the revised ‘Conclusions’ (Page 7: lines 240-245, in red), ten different metabolites were clearly mentioned as to whether those are ‘trans-cis’ isomers or a ‘trans’ isomer only.

(2) As per the reviewer’s helpful suggestion, thermodynamic stability study of the metabolites could be performed which will add more value to this paper. However, our present study is still in the stage of discovering novel derivatives, and it is unfortunate that we could not perform it as of now.

Some mistakes were found in the manuscript and were corrected in green, and ‘Abstract’ has been revised for clarification of wording. Revisions have also been made in the ‘Introduction’ (Page 1: lines 33-40) and in the ‘Conclusions’ (Page 7: lines 257-260) in red.
